# Food and Food Waste Antioxidants: Could They Be a Potent Defence against Parkinson’s Disease?

**DOI:** 10.3390/antiox13060645

**Published:** 2024-05-25

**Authors:** Claudia Cannas, Giada Lostia, Pier Andrea Serra, Alessandra Tiziana Peana, Rossana Migheli

**Affiliations:** Department of Medicine, Surgery and Pharmacy, University of Sassari, Viale San Pietro 43, 07100 Sassari, Italyapeana@uniss.it (A.T.P.)

**Keywords:** food, antioxidants, oxidative stress, neurodegeneration, Parkinson’s disease

## Abstract

Oxidative stress, an imbalance between reactive oxygen species (ROS) and endogenous antioxidants, plays an important role in the development of neurodegenerative diseases, including Parkinson’s. The human brain is vulnerable to oxidative stress because of the high rate of oxygen that it needs and the high levels of polyunsaturated fatty acids, which are substrates of lipid peroxidation. Natural antioxidants inhibit oxidation and reduce oxidative stress, preventing cancer, inflammation, and neurodegenerative disorders. Furthermore, in the literature, it is reported that antioxidants, due to their possible neuroprotective activity, may offer an interesting option for better symptom management, even Parkinson’s disease (PD). Natural antioxidants are usually found in several foods, such as fruits, vegetables, meat, fish, and oil, and in food wastes, such as seeds, peels, leaves, and skin. They can help the system of endogenous antioxidants, protect or repair cellular components from oxidative stress, and even halt lipid, protein, and DNA damage to neurons. This review will examine the extent of knowledge from the last ten years, about the neuroprotective potential effect of natural antioxidants present in food and food by-products, in in vivo and in vitro PD models. Additionally, this study will demonstrate that the pool of dietary antioxidants may be an important tool in the prevention of PD and an opportunity for cost savings in the public health area.

## 1. Introduction

Neurons are fundamental to the functioning of the human brain [1]; therefore, their progressive loss and death induce a process of neurodegeneration, which underlies the pathophysiology of many diseases [2]. Neurodegeneration includes several cell death pathways, such as autophagy [3], necrosis [4], and apoptosis [5]. Apoptosis is a process characterized by chromatin condensation, DNA fragmentation, and the activation of cysteine proteases and caspases [6]. Necrosis is a partially unknown mechanism, which is involved in several pathological conditions, such as hypoxia and hypoglycaemia, trauma, exposure to toxins, and, finally, ROS [4]. Lastly, autophagy is a common intracellular mechanism, able to remove misfolded proteins, a pathological feature of many neurodegenerative diseases [3]. Neurodegeneration is caused by several risk factors, among which are aging, genetic set up, and environmental aspects [1]. The risk factors mentioned above could induce a series of pathological mechanisms, such as protein aggregation, neuroinflammation, mitochondrial dysfunction, and oxidative stress [7].

Oxidative stress is a state of imbalance between ROS and endogenous antioxidant species [7]. Neuronal cells counteract oxidative stress via endogenous antioxidants able to neutralize ROS, limiting cell damage [8]. Free radicals can be generated by several physiological biochemical reactions, by drugs, toxins, and pollution [9]. The human brain is vulnerable to oxidative stress both because it requires high levels of oxygen and because high levels of polyunsaturated fatty acids, substrates of lipid peroxidation, are present [10,11].

Oxidative stress plays an important role in different pathologies in the central nervous system (CNS), such as Alzheimer’s disease, Huntington’s disease, amyotrophic lateral sclerosis, and Parkinson’s disease (PD). In the last few decades, PD has become the second most common widespread neurodegenerative disease, which affects 1–2% of the population over 65 [8]. PD is considered an α-synucleinopathy, because of the protein aggregation of misfolded α-synuclein and the formation of inclusions, called Lewy bodies, inside dopaminergic neurons. Additionally, PD is characterized by a progressive loss of dopaminergic neurons in the substantia nigra pars compacta (SNpc) [12]. Many studies affirm that the reduced activity in Complex I of the respiratory chain in the mitochondria of PD patients may contribute to an increase in ROS and, in turn, induce oxidative stress and apoptosis of neuronal cells [13]. The symptoms that characterize PD are mainly tremors, bradykinesia, muscle rigidity, and cognitive impairment. PD’s pathological mechanism is not completely known and, currently, there are no cures for this neurodegenerative disease, as therapies are limited to treating the symptoms and slowing down the progressive course [12]. Treating the symptoms of PD patients is highly expensive and entails economic impacts; in fact, the total direct and indirect health costs of Alzheimer’s, Parkinson’s, and all related disorders exceed hundreds of billions of euros per year across the European Union [14]. Most likely, the multifactorial nature of neurodegeneration on the basis of PD makes it difficult to find effective and efficient drug therapies. Researchers are looking to identify different pharmacological strategies to prevent and treat neurodegeneration. A valid option could be using bioactive and multi-target compounds already present in food and food wastes as a new avenue for the investigation of natural molecules with neuroprotective effects [7]. Natural molecules, as antioxidants, are compounds that inhibit oxidation, and it has been observed that they could reduce oxidative stress, preventing cancer, inflammation, and neurodegenerative disorders [15]. Therefore, antioxidants, due to their possible neuroprotective activity, may offer an interesting option for better managing PD [16].

Antioxidants are mainly classified into two types: synthetic antioxidants and natural antioxidants [17]. Natural antioxidants are generally categorized into endogenous and exogenous molecules (Figure 1). The exogenous ones are usually found in fruits, nuts, vegetables, other foods, and in a lot of food wastes, helping the system of endogenous antioxidants to protect or repair cellular components from oxidative stress and even to halt neuronal lipid, protein, and DNA damage [18].

This review will examine the neuroprotective potential effect of natural antioxidants present in food and food by-products, tested in the last ten years on in vivo and in vitro PD models. Additionally, this study will demonstrate that the pool of dietary antioxidants may be an important tool in the prevention of PD and an opportunity for cost savings in the public health area.

## 2. Material and Methods

The literature search was conducted on two scientific databases, including mainly articles published in the last ten years: PubMed and Scopus. The search strategy combined the descriptors using the Booleans operators (AND/OR) in this way: (Parkinson’s disease OR neurodegeneration) AND oxidative stress AND natural antioxidants AND (food OR diet) AND (food wastes OR food by-products). More than two hundred papers were retrieved. In this work, only papers precisely addressing the topics covered by the review were included. Each author independently conducted the selection, and eventual conflicts were resolved through confrontation and discussion.

## 3. Vitamins

### 3.1. Ascorbic Acid

Vitamin C, or ascorbic acid, an excellent exogenous and soluble-in-water antioxidant, plays a very important role in reductions in ROS in oxidative stress. Several foods are rich in vitamin C, such as many vegetables, citrus fruits, kiwifruits, and paprika [19]. Many works in the scientific literature state that the antioxidant activity of vitamin C, present in food and food wastes, could have important effects in the prevention of PD. The association between the consumption of vitamin C and the risk of development of PD was examined by Hughes et al. [20]. The authors analysed 1036 PD patients and their intake of antioxidant vitamins, such as vitamin C, administering a semi-quantitative food frequency questionnaire every four years. They observed, after the statistical analysis, a likely reduction in the risk of PD, because healthy patients have been associated with a higher dietary intake of ascorbic acid. Liu Y et al. [21] investigated the possible neuroprotective effect of ascorbic acid in a neuronal model of PC12 cells. They showed that ascorbic acid, used in different concentrations, is able to reduce colistin-induced oxidative stress and enhance superoxide dismutase (SOD) and glutathione (GSH) levels. In fact, the biosynthesis and activation of these antioxidant enzymes are stimulated by vitamin C, which leads to decreased intracellular ROS in cells [22]. Hantikainen et al. [23] presented a study, in which 43,865 men and women aged 18–94 years were followed from 1997 until 2016. Vitamin E, vitamin C, and beta-carotene intake was assessed via a detailed food frequency questionnaire. After a follow-up time of 17.6 years, 465 cases of PD were detected, evidencing that a higher vitamin E and vitamin C dietary intake was associated with a lower risk of PD. Bioactive compounds, like vitamin C, extracted from citrus wastes could be used for developing functional foods with anti-aging, anti-mutagenic, anti-oxidative, anti-inflammatory, and neuroprotective activities and are, therefore, exploitable from a pharmacological point of view for the treatment of several disorders, including neurodegenerative diseases [24]. In fact, in some studies, vitamin C and other antioxidants, contained in wasted parts of citrus fruits, were investigated by Sir Elkhatim et al. [25], in 2018. The work showed that peels of each type of citrus fruit had higher amounts of vitamin C and phenolic compounds than those in pulp and seeds. Moreover, among the citrus fruits analysed, orange peels contained the highest amount of vitamin C, compared to the peels of the other citrus fruits. In general, it can be said that the high quantity of antioxidants in citrus wastes, particularly the peels, indicate that they could be involved in the food or pharmaceutical industries for their affordability and, mainly, for their health benefits. An interesting experiment was conducted in 2020 on the development of dried snacks with a small size and low-weight kiwifruit, which are considered as unsaleable waste. Kiwifruits are an excellent source of vitamin C and other bioactive compounds, and the small waste ones could be a healthy ingredient for preparing dried wholesome snacks [26].

### 3.2. Vitamin E

“Vitamins E” is a term that includes tocopherol, tocochromanols, and tocotrienols. This vitamin represents an important fat-soluble antioxidant, able to counteract the lipid peroxidation by scavenging peroxyl radicals. Vitamin E also takes non-antioxidant activities. In fact, in addition to having a cytoprotective role for cellular components, it is vital in cellular signalling, such as the regulation of protein kinase C, cognitive functions, physical performance, and the regulation of gene expression. Vitamin E deficiency could lead to serious problems in humans, such as peripheral neuropathy, ataxia, and anaemia, because of the lack of its protection from free radicals, which damage the nerve cells [27,28,29]. Currently, the mechanisms of action of vitamin E are not completely known, but there is evidence about its neuroprotective effect. Matsura T. [30] investigated the protective effect of tocotrienol on PD in in vitro and in vivo models. SH-SY5Y cells, a neuroblastoma model, were treated with the toxin 1-methyl-4-phenylpyridinium (MPP+) and/or Tocotrienol (T3). It was observed that T3 significantly induced an improvement in cell viability, reduced by MPP+ in a dose-dependent way. Moreover, Matsura hypothesized that the protective effect of tocotrienol is dependent on several intracellular signal transduction pathways and independent of the antioxidant activity. Therefore, α-tocopherol could counteract the activation of extracellular signal-regulated kinase (ERK1/2), induced by hydrogen peroxide in PC12 cells. Zakharova et al. [31] suggest that the inhibition of ERK1/2 activity by α-tocopherol could reduce neuronal cell death in the brain under conditions of oxidative stress. Finally, a certain amount of vitamin E was observed, even among food wastes, such as grape seeds, which is a food by-product generated, in particular, in the wine industry. Grape seeds signify a small percentage of the total weight of the grape cluster. The great property of this by-product derives from its oil, which is a rich source of linoleic acid (60–70%) and tocopherols. It represents, with its antioxidant characteristics, mainly thanks to vitamin E, an important food by-product, which could benefit human health and should not be wasted [32]. High levels of α-tocopherol were found even in apple seeds, which were analysed by Akšić et al. [33] in Norway. In this study, the effective reuse of by-products from the apple industry was investigated, demonstrating that apple seeds are a good source for producing oil and a defatted oil cake rich in tocopherols. Apple seed oil and defatted oil cakes are important unconventional sources of beneficial health products.

### 3.3. Vitamin A and Carotenoids

In the diet, there are different types of vitamin A: (1) retinol and retinyl esters; (2) carotenoids, precursors of vitamin A; (3) non-provitamin A carotenoids. Vitamin A is a lipophilic vitamin, which is exclusively taken via diet. The main sources of vitamin A are liver, fish, eggs, and dairy products. On the contrary, carotenoids are abundant in orange and yellow vegetables and fruits [34].

Vitamin A and its active derivatives seem to have antioxidant and neuroprotective properties in many animal studies, and this is confirmed even by the important role of vitamin A and carotenoids during the development of the central nervous system and neuronal differentiation [35]. Some authors state that retinoic acid is essential in many signalling pathways and has also been implicated in the pathophysiology of Alzheimer’s disease and neurological diseases [36]. Retinoic acid-synthesizing enzymes are abundant in the mesotelencephalic dopamine system because dopaminergic neurotransmission is controlled by retinoid signal transduction, where retinoic acid is a crucial component. Therefore, it can be stated that retinoic acid and retinoic acid receptors are able to control the survival, adaptation, and homeostatic regulation of the dopaminergic system [35] and that several important homeostatic processes, such as inflammation, neuronal plasticity, and antioxidant activity, involve vitamin A and its derivatives. In the last 10 years, the role of vitamin A in neurodegenerative diseases has been largely investigated, and, in the literature, Marie et al. [37] studied the influence of vitamin A in the pathophysiology of PD. The authors assert that vitamin A and its bioactive derivative could be used for improving oxidative stress and for the survival of dopaminergic neurons in the SNpc. Different studies suggest the neuroprotective effects of vitamin A in PD. Marie et al. [38], in 2022, tested the influence of vitamin A addition in the diet in the dopaminergic neurons of a PD rat model. A 5-week preventive retinol supplementation was administered to adult male rats and motor and neurobiological lesions were subsequently induced by neurotoxin 6-hydroxydopamine (6-OHDA) unilateral injections in the striatum. The obtained results showed that the vitamin A supplementation improved voluntary movements and increased striatal dopamine levels in lesioned rats, in which an increased expression of dopamine receptors (D2) and retinoic acid receptors (isoform RXR) was observed. From this study, vitamin A supplementation seems to be a possible beneficial preventive approach for PD, improving striatal functions.

Additionally, Enenebeaku et al. [39] argue that agricultural food wastes, which are secondary products not converted into other functional forms, can be renewable and important resources of high nutrient levels, transformable into energy as well as animal feed. The authors’ findings revealed that agricultural wastes have high levels of many vitamins, showing an elevated level of vitamin A in cassava bagasse, a food waste produced by starch industries and rich in starch and fibre [40,41].

Carotenoids are tetraterpene pigments, belonging to the vitamin A family, which exhibit yellow, orange, red, and purple colours. Β-carotene, α-carotene, lycopene, β-cryptoxanthin, lutein, and zeaxanthin are mostly represented in food [42]. Several recent studies showed the neuroprotective role of carotenoids on neuronal in vitro and in vivo models. B-carotene, one of the most famous carotenoids, was investigated as a possible adjuvant in the therapy of PD by Chaves et al. [43]. B-carotene-loaded nanoparticles presented a significant neuroprotective effect against rotenone-induced damage in Drosophila melanogaster. Rotenone is a mitochondrial Complex I inhibitor, which has been demonstrated to be responsible for PD-like behavioural and neuropathological changes, such as the initiation of apoptosis and quickening of α-synuclein formation in PD models [44]. The researchers showed that β-carotene-loaded nanoparticles can protect against the locomotor damage and memory deficit induced by rotenone, ameliorating dopamine levels and oxidative stress [43]. Additionally, in 2013 a study was conducted by Chang et al. [45], which tested the neuroprotective effects of astaxanthin, beta-carotene, and canthaxanthin on undifferentiated rat pheochromocytoma (PC12) cells, a useful neuronal model for studying some neurodegenerative diseases. The co-authors demonstrated that astaxanthin, beta-carotene, and canthaxanthin have different grades of antioxidant activity in PC12 cells treated with amyloid beta-peptide (Ab(25-35)) and H_2_O_2_. The results demonstrated that astaxanthin has a more potent antioxidant activity than β-carotene in stabilizing free radicals, shown to be an all-round antioxidant. The astaxanthin revealed the strongest neuroprotective activity against Ab(25-35), fighting superoxide anion radicals, hydroxyl radicals, and H_2_O_2_, improving cell viability and reducing the production of ROS. Canthaxanthin inhibited only Ab(25-35)-induced cytotoxicity with partial neuroprotective activity. On the contrary, β-carotene showed lower scavenging and antioxidant activity, ranking as the lowest neuroprotectant in undifferentiated PC12 cells.

Another common carotenoid, abundant in fruits and vegetables, lutein, has been indicated to be a possible nutraceutical in the prevention of diabetic-related neurodegenerative diseases, because of its protective effects. A study, conducted by Chen et al. [46], showed that lutein significantly ameliorated the oxidative stress situation, reducing the amount of ROS in PC12 cells damaged by methylglyoxal (MGO), a cytotoxic by-product of glycolysis. This antioxidant seemed to reduce the apoptotic cell levels and mitochondrial damage, inhibiting the expression of caspase-3 and caspase-9 and through the PI3K/Akt signalling pathway.

Lycopene is a valid and anti-aging antioxidant. It was studied in combination with nicotinamide mononucleotide (NMN) by Liu X. et al. [47] on aging rats and senescent PC12 cells. Senescence was induced in both models by using D-galactose (D-gal), and the combination of lycopene and NMN showed better results than monotherapy in the prevention of aging. NMN and lycopene moderated the oxidative stress of rat brains and PC12 cells, inducing an increase in the endogenous antioxidant enzymes, such as superoxide dismutase (SOD), catalase (CAT), and glutathione peroxidase (GSH-Px), and reducing malondialdehyde (MDA) content. Additionally, both treatments significantly down-regulated the expressions of many p53, p21, and p16, which are senescence-related proteins with a pivotal role in the cell-intrinsic checkpoint, signalling, and repair responses. Lastly, lycopene and NMN led to the activation of Nrf2, which is a key transcription factor, responsible for reducing intracellular oxidative damage and delaying cellular senescence, in both aging models [48].

The unconsumed derivatives of food processing techniques contain a significant number of unutilized resources including carotenoids. In 2022, Dutta et al. [49] summarized several studies focused on the usage of different food wastes, like the pulp and peels of tomato, carrot, and sweet potato roots, rich in carotenoids. According to the researchers, carotenoids extracted from food wastes by green extraction technologies could be integrated into the daily diet.

### 3.4. Vitamin D

Vitamin D is considered an important neuroactive steroid, useful and fundamental for brain development and functions. This is demonstrated by the presence of vitamin D receptors and enzyme 1-alpha-hydroxylase, activators of vitamin D, in many areas of the human brain. Vitamin D may, therefore, be a good neurotrophic factor, protecting against excitotoxicity and enhancing other antioxidants [50]. It is crucial even for calcium homeostasis and the health of the skeleton and bones. This vitamin is mainly divided into two forms, which are present in nature: vitamin D2 (ergocalciferol) and vitamin D3 (cholecalciferol). Both formulae are found in food, such as almonds, walnuts, mushrooms, beans, green leafy vegetables, egg yolks, organ meats, and milk [51]. Cholecalciferol is also endogenously produced through the action of UV-B rays from the sun on the skin. The forms named above must be transformed into active forms by double hydroxylation for their functionality [28]. Vitamin D is transported as a pro-vitamin in the serum, bound to a vitamin D binding protein (DBP) to the liver, where it is hydroxylated to make 25-hydroxyvitamin D3 [25(OH)D3], the most circulating form of vitamin D. The blood concentration of this vitamin is a useful biomarker of an eventual vitamin D deficiency situation [52]. Some studies have reported a strong association between vitamin D deficiency and PD, and between a reduction in vitamin D and PD risk. Therefore, the development of PD and the loss of dopaminergic neurons could be characterized even by continual and insufficient quantities of vitamin D. Wang et al. [53], with a case–control study, evaluated the associations between serum vitamin D levels, vitamin D dietary intake, sunlight exposure, and newly diagnosed PD in a relatively large-scale Chinese population. The results of this study indicated that the mean levels of vitamin D serum seemed to be lower in patients with PD than normal cases, demonstrating that an increased risk for PD is significantly associated with lower levels of vitamin D serum and of sunlight exposure. On the other hand, a study, conducted by Liu Y. et al. [54], showed an inverse correlation between levels of vitamin D and the duration and the severity of PD. Therefore, lower vitamin D levels seem to correspond to a higher severity of PD in Chinese patients. Recent in vivo studies suggest that the function of vitamin D3 may be optimized by vitamin A co-administration. In line with the synergistic interplay between vitamins, Sirajo et al. [55] hypothesized that vitamin D3 could have potent slow-down effects on Parkinsonism in a haloperidol-induced mouse model of motor deficits when concomitantly administered with vitamin A. A significant decline in motor activity was registered in a PD mice model treated with haloperidol alone, compared to other experimental groups that received vitamin supplementations. Furthermore, the haloperidol-induced PD mice model treated with vitamin D3 and vitamin A showed a significant improvement in motor activity and an attenuation of oxidative stress levels and neurodegenerative features compared to other groups treated with vitamin A and vitamin D3 alone. Metal ions, such as manganese or zinc, at high concentrations, can usually cause oxidative stress and damage to mitochondria and cell membranes. Vitamin D3 could be able to reduce and ameliorate the levels of these ions, trough the activation of the SLC30A10 gene that expresses the zinc and manganese transporter, helping to transport them away [56]. SLC30A10 is an essential gene for zinc and manganese homeostasis, and mutations in this gene can result in impaired function in the zinc and manganese transporter, leading to manganese intoxication, with Parkinson-like symptoms [57]. Ultimately, vitamin D is present even in food wastes, such as fishery wastes [58,59]. In the world, there is a high production of fishery products and huge quantities of fishery residues, such as viscera, carcass, head, skin, or bones. Fishery residues represent a high percentage, and from this, only about 30% is recycled [60]. Given the high percentage of fishery residues, a bigger effort should be made to upgrade this class of by-products, using them as a substitute for protein in animal feed and as a source of functional products [32].

### 3.5. Vitamins B

Thiamine (vitamin B1), riboflavin (vitamin B2), niacin (vitamin B3), pantothenic acid (vitamin B5), pyridoxine (vitamin B6), biotin (vitamin B7), folate (vitamin B9), and cobalamin (vitamin B12) belong to the B water-soluble family of vitamins. The B vitamin functions are the regulation of metabolism, improvement in the immune and nervous system, and the promotion cell growth and cell division. Vitamins B are enzyme cofactors in multiple biochemical pathways inside the organism [28].

#### 3.5.1. Vitamin B1 (Thiamine)

Humans need a continuous supply of thiamine in their diet; on the contrary, fungi, plants, and most bacteria are able to synthesise thiamine on their own. There are several nutritional sources of thiamine, such as whole grains, bread, pork meat, legumes, and nuts. Vegetables and fruits are less rich in vitamin B [61].

Vitamin B or thiamine deficiency seems to be related to a higher risk of PD. In fact, Håglin et al. [62] stated that a low dietary intake of thiamine in the 8 years before PD diagnosis is related to olfactory dysfunction, which is one of the non-motor symptoms associated with a major risk of PD. Therefore, it is important to assume the correct quantities of vitamin B1 because thiamine deficiency supports neuronal death and increases the possibility of developing PD. Moreover, thiamine supplementation may restore pathological changes and improve motor and non-motor symptoms associated with PD. Additionally, a thiamine decrease is correlated with a reduction in the α-ketoglutarate dehydrogenase complex, an enzyme associated with thiamine deficiency that could aggravate neurodegeneration in PD patients [63]. These findings are supported by Costantini et al., in a clinical study on PD patients. The coauthors state that PD patients, treated with high doses of thiamine and no other anti-Parkinson therapy, improved their motor symptoms from 31.3% to 77.3% [64]. It can be affirmed that lower thiamine levels are related to neurodegeneration and the aging process [65].

Among agrifood by-products, milk serum contains 0.38 mg/mL of thiamine. Torres-Leòn and his coauthors suggest the use of the nutraceuticals of milk serum in the production of dietary supplements [32].

#### 3.5.2. Vitamin B3

Vitamin B3, also called niacin, can be synthesized by plants, most fungi, and bacteria. Additionally, most mammals, including humans, are able to produce nicotinamide, principally in the liver, by using the essential amino acid tryptophan. For this reason, tryptophan can be considered one of the sources of niacin, in addition to the direct dietary sources. Since the conversion from tryptophan to niacin cannot cover the needs of niacin, it is necessary to consume food, such as meat, whole grains, milk, and dairy products. Peanuts, fish, mushrooms, yeasts, legumes, and nuts are also suitable sources of niacin. Regular coffee consumption conspicuously contributes to niacin intake in humans, as well [61].

Niacin needs to be activated in nicotinamide (the active form), which is an essential precursor of nicotinamide adenine dinucleotide (NADH) and nicotinamide adenine dinucleotide phosphate (NADPH). NADH and NADPH are coenzymes implicated in over 200 enzymatic reactions in the organism, such as the production of adenosine triphosphate (ATP) [28]. NAD was demonstrated to be a neuroprotective antioxidant by Lu L. et al. [66], who stated that NMN plays significant roles in a variety of biological processes, including energy metabolism, mitochondrial functions, calcium homeostasis, and anti-oxidation/generation of oxidative stress. PC12 cells, in a neuronal model, were treated with rotenone. Additionally, several concentrations of NMN were tested, and, later, cell survival, apoptosis, necrosis, NAD+, and ATP levels were investigated. NMN could restore the cell viability and reinstate intracellular levels of NAD+ and ATP in rotenone-compromised PC12 cells. Thus, the researcher’s hypothesis suggests that NMN could decrease PD-like pathological changes, attenuating apoptosis and improving energy metabolism in an in vitro model.

Several studies showed that ATP depletion, mitochondrial dysfunction, and oxidative stress play an important role during neuronal death in PD. Motawi et al. demonstrated that GSH, SOD, and ATP depletion, induced by rotenone, improve after niacin administration with levodopa, which induce a decrease in the free radical levels associated with PD [67].

#### 3.5.3. Vitamin B6

Vitamin B6 or pyridoxine is an exogenous, from dietary sources, or an endogenous, from the microbiota of the human intestine, cofactor. Pyridoxine and all its derivatives are present in a series of vital metabolic processes, such as amino acid metabolism and synthesis of proteins, polyamines, lipids, and carbohydrates. Additionally, it has mitochondrial functions and acts as a neurotransmitter and antioxidant [63,68].

Animal-based foods, such as meat, milk, and certain fish, are richer in vitamin B6 than plant-based foods, with the exception of cabbage, potatoes, beans, whole grains, peanuts, and soybeans. Three forms of vitamin B6 exist in nature: pyridoxine (PN), pyridoxal (PL), and pyridoxamine (PM) [69].

The functional form of vitamin B6, which is pyridoxal 50-phosphate, and its severe deficiency could have a relation with many neurological disorders, such as PD [63]. Pyridoxine is a great antioxidant vitamin, with the ability to remove singlet oxygen as the other antioxidant vitamins. Shen L. et al. [70] reported that an insufficient quantity of vitamin B6 can lead to oxidative stress in rats’ organs; for this reason, its supplementary administration can reduce oxidative stress in the cells. In conclusion, these findings can support the hypothesis that the antioxidant potential of vitamin B6 may help to prevent PD via the inhibition of oxidative stress. Furthermore, Hikal and co-authors point to banana peel, containing vitamin B6, as an important waste treasure and source of natural antioxidants that prevent oxidative stress [71].

#### 3.5.4. Vitamin B12

Vitamin B12, known even as cobalamin, is an essential micronutrient involved in the growth of the nervous system, and, as reported by Jia L. [72], many brain disorders, memory decline, acute psychosis, and similar problems are related to low levels of vitamin B12. This work corroborates the hypothesis that vitamin B12 could effectively improve the cognitive function of model animals and, consequently, of patients.

The richest foods in vitamin B12 are milk products, meat, and fish [63]. On the contrary, plant-derived foods usually do not contain important vitamin B12 amounts, except dried green and purple lavers [73]. Several studies have focused attention on the determination of the correlation between the deficiency of vitamin B12 and PD, reporting that deficiency of vitamin B12 in PD patients is associated with cognitive impairment, aggravation of the disease with neuropathy, and motor symptoms [63].

McCarter et al. [74] studied the correlation between vitamin B12 in patients’ serum at the time of PD diagnosis and dementia risk, utilizing a population of Parkinsonian patients. The results of this study show a higher level of vitamin B12 in the serum of PD patients without dementia than patients with dementia, suggesting that higher levels of serum vitamin B12 at PD diagnosis are correlated with a lower risk of future dementia. According to Wu Y et al. [75], vitamin B12 could be a coadjutant for the treatment of PD, because it seems to ameliorate in vitro and in vivo PD models. A pretreatment with vitamin B12 significantly reduced rotenone-induced cell damage in the SH-SY5Y cellular model, improving the oxidative stress situation and reducing the quantity of ROS through the mediation of oxidative-stress-related proteins. Subsequently, the antioxidant effects of vitamin B12 were examined in vivo, using two models: C. elegans and mouse models. The vitamin B12 treatment in the Parkin gene knockout C. elegans PD model significantly reduced the intensity of the ROS fluorescence signal and alleviated the motor deficits in C. elegans. Similarly, vitamin B12 treatment alleviated movement deficits and the degeneration of DA neurons in the 1-methyl-4-phenyl-1,2,3,6-tetrahydropyridine (MPTP)-induced mouse PD model. MPTP is known to lead to a selective loss of dopaminergic neurons and a consequent dopamine reduction in the SNpc of models [76].

## 4. Polyphenols

Polyphenols are natural molecules produced by the secondary metabolism of plants. They seem to have potential therapeutic effects in the context of many diseases, and, for this reason, they have captured the attention of researchers. Polyphenols have the ability to inhibit free radicals and, moreover, activate antioxidant enzymes, reducing the imbalance between ROS and endogenous antioxidants. Therefore, these natural compounds demonstrate interesting effects in the prevention of several diseases, such as diabetes, obesity, cancer, cardiovascular diseases, osteoporosis, and neurodegenerative diseases, gaining growing attention in the scientific world [77]. High concentrations of polyphenols can be taken from leafy vegetables, citrus fruits, red wine, berries, olive oil, and tea [78]. Polyphenols are divided into flavonoids and non-flavonoids on the basis of their chemical structure [79]. Flavonoids are one of the largest families of phenolic compounds, found in many foods. They are divided into six major subclasses: flavones, flavonols, flavanones, flavanols, isoflavones, and anthocyanins [80].

Catechins and theaflavins are polyphenolic compounds, belonging to the flavonoid family and to the subclass of flavanols. They are present in many categories of foods and drinks, such as green tea, wine, cocoa-based products, and black tea. Catechins and theaflavins possess antioxidant properties and could have beneficial effects in the prevention and protection against diseases caused by oxidative stress [81,82], such as PD. Catechins are the most important and well-preserved polyphenols present in green tea leaves. Theaflavins are more represented polyphenols in black tea, obtained by a fermentation of fresh green tea leaves. Zhang et al. [83] confirmed, in their study, that the quantity of tea catechins and theaflavins could be affected by many factors, such as the geographical location, leaf grade, and leaf size. These results could be a good indication to evaluate the antioxidant capacity of teas, based on the characteristics mentioned above. Lee et al. [84], in 2022, evaluated the neuroprotective effect in different degrees of green tea fermentation. The authors demonstrated that despite the lower catechin content in favour of increased theaflavins of semi-fermented and fermented tea, both are able to reduce MPTP-induced neurotoxicity in in vitro and in vivo damage models, SH-SY5Y cells, and mice. Therefore, fermented tea could be considered to have a neuroprotective effect in neuro-oxidative disorders, including PD.

Another flavonoid abundant in nature is quercetin (QCT), belonging to the flavanol group, which exists in various vegetables and fruits, such as apples, berries, cherries, red leaf lettuce, onions, and asparagus [85]. In a recent study, conducted in 2022 by Lin ZH et al. [86], QCT could protect in vitro and in vivo PD models against MPTP. The results showed that QCT significantly inhibited ferroptosis, a form of programmed cell death characterized by iron-dependent lipid peroxidation, in M17 and PC12 cells, activating the essential nuclear factor erythroid 2-related factor 2 (Nrf2) protein. This protein has anti-inflammatory and antioxidant functions. Moreover, QCT ameliorated motor behavioural deficits and protected against the loss of dopaminergic neurons in MPTP-induced PD mice models, demonstrating that QCT could be a potential neuroprotective agent for preventing neurodegenerative disease.

The flavonol group is characterized by isoquercitrin, a natural molecule, similar to quercetin, which is a bioactive ingredient of agricultural waste apple pomace. It is known for its anti-oxidation, anti-aging, and anti-inflammation properties. In a study conducted by Liu C. et al. [87], this flavonol was tested in MPTP-induced PD mouse models. The results obtained after the isoquercitrin treatment showed an amelioration of the animal behaviours against MPTP-induced neurotoxicity and a reduction in the loss of dopamine neurons induced by MPTP. Additionally, dopamine transporter expression was increased, and the pro-apoptotic signalling molecules and MPTP-triggered oxidative stress were reduced by isoquercitrin, demonstrating its protective effects.

Phenolic acids belong to non-flavonoids and are present in acidic-tasting fruits and in vegetal substances, constituting a third of the polyphenolic compounds in our nutrition [88]. Long et al. [89] provided novel insights for ferulic acid (FA), a known phenolic acid. They demonstrated that it could be a useful tool of prevention for neurodegenerative diseases such as PD. The FA anti-oxidative activity was tested on C. elegans nematodes, an in vivo model, H2O2, and 6-OHDA treated. The results suggested that FA can reduce ROS levels, improving the neurodegeneration in both PD models. Additionally, in this study, the authors showed that FA induced a reduction in α-synuclein accumulation, improving motility, demonstrating an increase in the survival rate of C. elegans. Similar results were obtained in in vitro experiments, where the role of ferulic acid has been investigated in 6-OHDA- or H2O2-treated PC-12 cells. They discovered that FA improves cell viability and reduces the ROS levels in damaged PC-12 cells, even by inhibiting apoptosis, once again proving the protective effect of FA.

Many agrifood by-products are rich in bioactive substances, such as phenolic acids. The residue in olive pomace skin, generated in olive oil, is plentiful in these substances, and this was confirmed by Romero et al. [90]. The researchers examined the concentration of bioactive substances in olive oil and olive by-products. The olive wastes included leaves and pieces of olive skin, together with small pieces of pulp, coming from the cleaning of the pit fragments present in the olive pomace. Romero et al. [90] state that the olive pomace skin waste product is a source of maslinic acid, a non-flavonoid among the triterpene acids and known for its antioxidant activity. Maslinic acid has really caught the attention of researchers for its variety of biological activities, such as anti-tumour, anti-inflammatory, antioxidant, and even neuroprotective activities. Last year, a study was conducted by Cao et al. [91] about the potential neuroprotective effect of maslinic acid on PD mice models. The results showed that the oral administration of maslinic acid on PD mice with locomotor deficits improved the motor functions. Furthermore, it was demonstrated that low and high doses of maslinic acid prevented dopaminergic neuronal loss in PD mice.

Stilbenes, another group of non-flavonoids, are characterized by two phenyl moieties connected by a two-carbon methylene bridge. One of the most extensively studied and famous stilbenes is resveratrol. Resveratrol, in which red wine is rich, is known for having many pharmacologic properties, including cardioprotection, anti-oxidation, anti-cancer, anti-inflammatory, and neuroprotective effects. In a study conducted by Lofrumento et al. [92], twenty-four adult male MPTP-treated mice were subjected to daily resveratrol administration for 3 weeks. It was observed that resveratrol induced important neuroprotection of dopaminergic neurons against MPTP-induced neurotoxicity. Moreover, resveratrol led to a down-regulation of the pro-inflammatory IL-1β, TNF-α, and IL-6 cytokines and their corresponding cytokine receptors. Similar results were obtained by Gaballah et al. [93], in 2016. In fact, they demonstrated the potential effects of resveratrol on endoplasmic reticulum stress-associated apoptosis and on oxidative and inflammatory markers. Resveratrol improved rotenone-treated rats’ behaviour and significantly increased striatal dopamine levels. Regarding oxidative and inflammatory markers, this stilbene decreased striatal IL-1 levels and suppressed oxidative stress markers in rotenone-treated rats. Other kinds of non-flavonoids, such as sesamin and sesamolin, are even contained in Sesamum indicum oil. Sesamin and sesamolin are lignans and they were tested in a neuronal PC12 cell model by Ramanzani et al. [94] in 2023. These diphenolic compounds are known to possess anti-inflammatory, antioxidant, and anti-apoptotic properties. Their neuroprotective effect was evaluated in 6-OHDA-treated PC12 cells. It emerged that the cells pre-treated with sesame seed extracts significantly improved cell viability and reduced ROS and 6-OHDA-induced apoptosis.

*Vitis vinifera* red grape is another possible source of antioxidants, and, more precisely, its wastes, such as seeds, leaves, the pulp, and the skin extract, are a source of many phenolic compounds. Flavonoids, anthocyanins, catechins, and stilbenes, present in red grape wastes, have beneficial effects against oxidative injury in different organs, such as the kidneys, liver, heart, and brain. The neuroprotective effect of Vitis vinifera red grape seed and skin extract was investigated in both in vitro and in vivo PD models by Ben Youssef et al. [95] in 2021. The researchers used the neurotoxin 6-OHDA to induce oxidative damage and the degeneration of dopaminergic neurons. The seeds and skin extracts induced a decrease in apoptosis and ROS levels in 6-OHDA PD models. In addition, they were able to protect against neuronal dopaminergic loss, improving the neurodegeneration on the basis of PD.

## 5. Trace Elements

### 5.1. Selenium

Selenium (Se) is an essential micronutrient and a cofactor of many biological enzymes, such as glutathione peroxidase. It has important protective properties against oxidative stress. It is considered a free radical scavenger of fundamental importance for human health, and, for this reason, a proper dietary intake of this essential trace element may be a solution to reduce extreme selenium deficiencies and, consequently, to reduce the risk of developing related chronic degenerative diseases [96]. The main food sources of selenium in the diet are animal-based foods (meat, fish, milks, and eggs), cereals, vegetables, and fruits. Usually, vegetables have a small amount of Se, except cruciferous vegetables, garlic, and onions, which are considered high-Se-accumulating vegetables. Additionally, nuts are one of the ten foods with the highest selenium content [97]. The protective activities of selenium against drugs, toxic heavy metals, carcinogens, mycotoxins, and pesticides have been proved. In fact, in mammal organisms, there is a large number of selenoproteins, such as glutathione peroxidase and thioredoxin reductase, which play an important role as antioxidant enzymes [98]. Liu W et al. [99] confirmed that selenium could have a neuroprotective role in oxidative stress and the mitochondrial dysfunction of PD, contributing to slowing down PD progression. The authors tested different concentrations of selenium in MPP+-induced PD in an in vitro model (PC12 cells). Se significantly reduced MPP+-induced cell damage in PC12 cells, demonstrating that it could be a neuroprotector.

### 5.2. Zinc

Zinc (Zn) is another element of living beings, essential in many biological processes and in the proper growth of plants, animals, and humans [100]. Additionally, Zn is considered an antioxidant, implicated in many mechanisms, which decreases the generation of ROS. Zn acts as a co-factor of the enzyme superoxide dismutase (SOD) and as a competitor of Fe^2+^ and Cu^2+^ ions, displacing these redox-active metals. Lastly, Zn activates endogenous antioxidants, such as glutathione (GSH), catalase, and SOD, and inhibits oxidant-promoting enzymes such as inducible nitric acid synthase (iNos) and NADPH oxidase [101]. Therefore, zinc deficiency, related to low diet quality, could be implicated in PD, as stated by Mbiydzenyuy et al. [102], which affirmed that PD patients present low levels of zinc in cerebrospinal fluid. The authors evaluated the role of Zn and linoleic acid co-administration. The neuroprotective effect was evaluated in rotenone-induced PD murine models. Interestingly, pre-treatment of rats with zinc and linoleic acid improved the rotenone-induced neuronal loss in the midbrain and prevented the reduction in antioxidant activity of endogenous antioxidants. According to the authors, the antioxidant effect of zinc could be related to its collaboration with metallothioneins, which are rich in cysteine and excellent scavengers of free radicals.

## 6. Other Antioxidants

### 6.1. Linalool

Linalool is a monoterpene molecule, known for its anti-inflammatory and antioxidant properties, abundant in many essential oils from aromatic plants, such as lavender (*Lavandula angustifolia* Mill.), bergamot (*Citrus bergamia* Risso), and orange (*Citrus sinensis* L.) [103]. Additionally, this molecule is largely used in food additives [104]. Many authors state that linalool possesses neuroprotective effects in several cerebral ischemia models, decreases oxidative stress induced by oxygen-glucose deprivation, and is a scavenger of peroxyl radicals [105]. In a recent study, Migheli et al. [106] proved and confirmed that linalool may be a good adjuvant for preventing neurodegenerative diseases, such as PD. In fact, the researchers tested this compound in a neuronal model, PC12 cells, demonstrating a reduction in the H_2_O_2_-induced damage by linalool. According to the authors, linalool reduced LDH release, ROS, and apoptosis levels in PC12 cells damaged with H_2_O_2_. In addition, discrete percentages of linalool are present even in food wastes, as confirmed by Geraci et al. [107], who quantitatively and qualitatively assessed the essential oil components of various orange peels.

### 6.2. Caffeine

Caffeine (1,3,7-trimethylxanthin) is a natural molecule, belonging to the class of alkaloids. Coffee, tea, and cocoa plants are rich in this psychostimulant, which is able to prevent lipid peroxidation and the production of ROS. Therefore, a regular consumption of caffeine could improve the mitochondrial functions and oxidative stress, reducing neurotoxicity in PD patients [108]. Xu et al. [109] found confirmation of the neuroprotective role of caffeine in PD MPTP-induced models. The results of their study reported a significant attenuation of MPTP-induced dopamine depletion in mice by caffeine, but the authors confirmed the dependence of caffeine on the adenosine A2AR receptors on forebrain neurons to perform its neuroprotective and locomotor activating properties. On the contrary, caffeine is totally independent of A2Ars receptors in astrocytes to perform its protection on dopaminergic neurons.

### 6.3. Zingerone

Ginger is a popular rhizome, belonging to the Zingiberaceae family, saleable as fresh root, syrup, brine, or dried ginger spice. Zingerone, chemically called 4-(4-hydroxy-3-methoxyphenyl)-butan-2-one, is subsequently obtained in the major part, from the drying or cooking of this rhizome. The pharmaceutical properties of zingerone are well known. In fact, Angelopoulou et al. [110] reported, in their review, its antioxidant properties and capacities to improve the endogenous antioxidant mechanisms. Zingerone is able to restore the activity of the antioxidant enzymes, such as SOD and catalase. Additionally, the authors reported the hypothesis of the amelioration of levodopa-induced dyskinesias and oxidative stress in PD by zingerone, which was also confirmed by Choi et al. [111], in 2015. They examined the effect of zingerone on neuronal cell death in a MPTP-mediated PD model, suggesting possible neuroprotective activity against PD. Zingerone induced the activation of the ERK pathway and VMAT2 expression [110,111]. ERK, the extracellular signal-regulated kinase, is implicated in the activity of the VMAT2 promoter, which expresses the vesicular monoamine transporter 2 (VMAT2) [110]. Pifl et al. [112] state that the function of VMAT2 is reduced in PD patients, proposing the hypothesis that VMAT2 defects could promote mechanisms leading to nigrostriatal DA neuron death in PD. For this reason, zingerone may be a valid tool for an increase in ERK activation, which, consequently, could lead to increased VMAT2 expression in PD [110].

## 7. Discussions

In recent years, the attention of researchers has been focused on food and food wastes as sources of antioxidant compounds. Natural antioxidants, such as glutathione, vitamins, carotenoids, and polyphenols, via food or food wastes, could be considered to prevent the oxidative stress caused by free radicals in dopaminergic neurons [27]. The potential antioxidant effect of several dietary components and agrifood by-products may prevent cognitive decline. This hypothesis has been shown in different studies, where it was established that a combination of several nutrients can have better effects than those attributable to individual nutrients. It can be said with certainty that lifestyle and diet are evidently fundamental for cognitive functions [113,114].

In fact, as shown in Table 1, several exogenous antioxidants, present both in food and food wastes, can also be found in human blood and the brain at different concentrations.

Thus, it is plausible that the complex arrangement of nutrients that characterizes a dietary pattern induces synergistic effects more effectively than single nutritional components in relation to neurodegeneration [129].

Other studies showed that the observance of a healthy diet, such as the Mediterranean diet, is associated with a lower probability of prodromal PD in older people. This study confirms that the potential and interactive effects of different foods and nutrients that characterize a dietary pattern could delay the onset or lower the incidence of PD [130]. Several molecular mechanisms have been hypothesized to be on the basis of neurodegenerative diseases. Oxidative stress is one of them and is reduced by bioactive antioxidant components of food, present in a healthy diet [129]. Recently, several studies have shown that antioxidants are also present in food wastes. Food wastes and agrifood by-products have precious components with antioxidant activities, which could be introduced into the diet (Table 2).

As reported in Table 2, food and food wastes present similar quantities of antioxidants, proving that they could be analogous neuroprotective sources. The usually non-edible portions of fruits, such as peels or seeds, contain high amounts of bioactive compounds and, sometimes, even higher than the portions that are normally eaten. The valorisation of food by-products into edible materials could be an interesting approach in this field [113,137,138,139]. Our aim is to consider food and food waste as a source of compounds with antioxidant activity, capable of preventing PD. Vitamins, such as ascorbic acid, tocopherol, and vitamins A, B, and D, could be a valid tool to reduce oxidative stress and damage in neuronal PD models. Polyphenols, such as flavonoids, are bioactive molecules, present in food, that are able to stabilize free radicals and, moreover, activate endogenous antioxidant enzymes, as demonstrated by Zhang et al. [83] and Lee Y.R. et al. [84]. Non-flavonoids polyphenols, like phenolic acids, abundant in aliments, such as blackberries, grapes, ginger, and rhubarb, have been tested in in vivo and in vitro models. Long et al. [89] demonstrated the antioxidant and neuroprotective effect of ferulic acid in PD models.

Selenium and zinc, trace elements present in cereals, meat, dairy products, fish, mushrooms, garlic, and asparagus, have shown antioxidant capacity, as reported by Liu and coworkers [99] and Mbiydzenyuy and colleagues [102]. Neuroprotective effects in PD models were found even in other antioxidants, such as caffeine, zingerone, and linalool, which have in common the ability to reduce free radicals and are implicated in specific signalling pathways [106].

## 8. Conclusions

This analysis leads us to hypothesize that food and food wastes should be considered as prophylaxis and could be the solution to reduce or prevent oxidative-stress-mediated pathologies that majorly impact our era, such as PD. Further investigations are necessary to evaluate the possibility of feedback on the social, economic, and health impacts on the prevention of PD, which is very widespread throughout the world. This review examined the extent of knowledge from the last ten years, about the neuroprotective potential effect of natural antioxidants, present in food and food by-products, in in vivo and in vitro PD models. Additionally, this study suggests that the pool of dietary antioxidants may be an important tool in the prevention of PD and an opportunity for cost savings in the public health area.

## Figures and Tables

**Figure 1 antioxidants-13-00645-f001:**
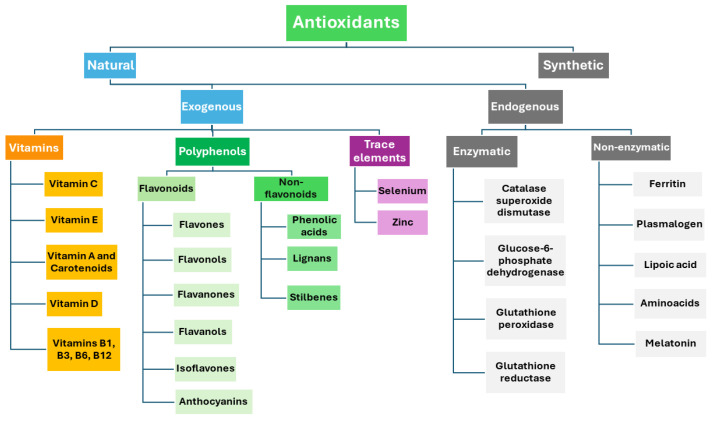
Antioxidant classification.

**Table 1 antioxidants-13-00645-t001:** Exogenous antioxidants’ concentrations in blood and brain.

Antioxidants	Exogenous Antioxidants’ Levels in Blood	Exogenous Antioxidants’ Levels in Brain	Ref.
Vitamin C	0.977 ± 0.086 ng/μL	1–10 mM	[115,116]
Vitamin E	(α-tocopherol) 34.9 ± 2.84 µmol/L	(Tocopherols isomers) 0.11–17.9 nmol/g	[117,118]
Vitamin A	0.3–0.6 mg/L	87.8–163.3 pmol/g	[118,119]
Carotenoids	Β-carotene 0.327 ± 0.054	1.8–23.0 pmol/g	[117,118]
Vitamin D	50 nmol/L (20 ng/mL)	Not found	[120]
Thiamine (B1)	679.3 ± 188.7 ng/μL	Not found	[115]
Niacin (B3)	0.260 ± 0.053 ng/μL	0.5 mmol/kg	[115,121]
Pyridoxine (B6)	0.368 ± 0.201 ng/μL	Not found	[115]
Vitamin B12	0.405 ± 0.291 ng/μL	0.027 ± 0.013 pg/g wet tissues	[115,122]
Polyphenols	171.6 ± 3.1–208.4 ± 2.45 (mg L^−1^) GAE	1 nmol/g of brain tissue	[123,124]
Selenium	60–150 ng/mL	552–1435 ng/g	[125,126]
Zinc	585.2–607.0 µg/100 mL	150 µmol/L	[127,128]

**Table 2 antioxidants-13-00645-t002:** Food and food waste antioxidants.

Antioxidant	Food	Antioxidant Content in Food	Food Waste	Antioxidant Content in Food Waste	Ref.
Ascorbic acid (mg/100g)	Orange	82.34	Peel	110.4	[25,131,132]
	Lemon	58.12	Peel	58.59	
	Tomato	5.71–101.29	Peel	110.00	
			Seeds	9.50	
Vitamin A	Potato	91 μg/g dry weight	Peel	0.24 ± 0.03 mg/L	[39,133]
Tocopherols	Apple	4.54–18.56 mg/100 g dry weight	Seeds	1.391–1.811 µg/g	[134]
Vitamin B3/Niacin	Potato	1035–1573 μg/100 g	Peel	3.25 ± 0.20 mg/L	[39,61]
Vitamin B6	Potato	0.16 mg/100 g	Peel	0.55 ± 0.21 mg/L	[39,69]
Polyphenols	Strawberry	422 ± 15 mg/100 mg of fresh mass	Pomace	24.4 mg gallic acid/g dry sample	[135,136]

## Data Availability

Not applicable.

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
