# Peer review of "Food and Food Waste Antioxidants: Could They Be a Potent Defence against Parkinson’s Disease?"

_antioxidants, 2024, doi:10.3390/antiox13060645_

Round 1

Reviewer 1 Report

Please correct the title, food wastes does not sound right.

Material and methods section is missing.

Please indicate a division of the material into food and food waste for a easier understanding.

Discussions and conclusions should be separated and a more comprehensive part should belong to discussion.

Figure 1 needs a higher resolution, the words written black on gray are hard to distinguish

Although this manuscript is a review, a minimal description of how the included papers were selected should be formulated as a material and methods section.

 I suggest a separation of the material into food and food by-products antioxidants. 

Discussions should be distinct from conclusions.

English language needs some corrections.

Reviewer 2 Report

antioxidants 3004281

Food and food wastes antioxidants: Could they be a potent defence against Parkinson’s disease?

In this review, the authors examine the extent of knowledge over the past ten years on the potential neuroprotective effect of natural antioxidants, present in foods and food by-products in vivo and in vitro models of Parkinson's disease.

Overall, the review is well structured, although the topic is not too original. To enhance the novelty and interest of the manuscript and, in order to make it more interesting, some additional figures should be included. One of them could be to provide a Diagram or Table mentioning the concentrations of exogenous and endogenous antioxidants, in blood and brain (when known), which provide protective or beneficial effects against Parkinson's disease. This aspect would allow possible interventions using antioxidant dietary supplements to be more critically evaluated.

The title with a question mark could be changed to an affirmative phrase.

Round 2

Reviewer 1 Report

The revised form of the manuscript has been significantly improved.

The authors rearranged the sections as indicated and improved the appearance and the content of the manuscript.

Reviewer 2 Report

antioxidants 3004281

Food and food wastes antioxidants: Could they be a potent defence against Parkinson’s disease?

In this review, the authors examine the extent of knowledge over the past ten years on the potential neuroprotective effect of natural antioxidants, present in foods and food by-products in vivo and in vitro models of Parkinson's disease.

The manuscript is fine in the present form.

The manuscript is fine in the present form.
